# Enhancing Border Gateway Protocol Security Using Public Blockchain

**DOI:** 10.3390/s20164482

**Published:** 2020-08-11

**Authors:** Lukas Mastilak, Marek Galinski, Pavol Helebrandt, Ivan Kotuliak, Michal Ries

**Affiliations:** Faculty of Informatics and Information Technologies, Slovak University of Technology in Bratislava, Ilkovicova 2, 842 16 Bratislava, Slovakia; marek.galinski@stuba.sk (M.G.); pavol.helebrandt@stuba.sk (P.H.); ivan.kotuliak@stuba.sk (I.K.); michal.ries@stuba.sk (M.R.)

**Keywords:** BGP hijacking, BGP security, blockchain, management, network

## Abstract

Communication on the Internet consisting of a massive number of Autonomous Systems (AS) depends on routing based on Border Gateway Protocol (BGP). Routers generally trust the veracity of information in BGP updates from their neighbors, as with many other routing protocols. However, this trust leaves the whole system vulnerable to multiple attacks, such as BGP hijacking. Several solutions have been proposed to increase the security of BGP routing protocol, most based on centralized Public Key Infrastructure, but their adoption has been relatively slow. Additionally, these solutions are open to attack on this centralized system. Decentralized alternatives utilizing blockchain to validate BGP updates have recently been proposed. The distributed nature of blockchain and its trustless environment increase the overall system security and conform to the distributed character of the BGP. All of the techniques based on blockchain concentrate on inspecting incoming BGP updates only. In this paper, we improve on these by modifying an existing architecture for the management of network devices. The original architecture adopted a private blockchain implementation of HyperLedger. On the other hand, we use the public blockchain Ethereum, more specifically the Ropsten testing environment. Our solution provides a module design for the management of AS border routers. It enables verification of the prefixes even before any router sends BGP updates announcing them. Thus, we eliminate fraudulent BGP origin announcements from the AS deploying our solution. Furthermore, blockchain provides storage options for configurations of edge routers and keeps the irrefutable history of all changes. We can analyze router settings history to detect whether the router advertised incorrect information, when and for how long.

## 1. Introduction

Nowadays, the Internet is composed of a huge number of Autonomous Systems (ASes). These ASes exchange information about IP prefixes. Communication among ASes is handled using a routing protocol called the Border Gateway Protocol (BGP) [1]. In general, ASes trust each other and assume that their neighbour advertises correct data. This behaviour is a little naive because accepting all incoming BGP advertisements as genuine and correct may open the door for potential attacks.

One of the most widespread types of attacks on BGP is called BGP hijacking [2]. Its objective is to steal the network IP address and reroute traffic intended for this network by an Autonomous System (AS) under control of a malicious actor. The router of malicious AS announces network prefixes that do not belong to the attacker by inserting false information into BGP messages sent to the neighbouring ASes. Routers in peer ASes accept the advertised prefixes and modify how traffic destined for the hijacked networks is routed. The prefix advertisements are propagated further to their neighbouring ASes, and this process is repeated throughout the Internet. Every AS updates its routing table according to fraudulent values in the messages received from neighbours. When the network converges, all traffic hijacked from the originating AS is diverted to the malicious AS. This type of attack has a few variations depending on the hijacker’s aims. The attack can propagate IP prefix with the same mask as was the prefix in the original AS, but with better features for the prefix. Using another method, the prefix can be advertised with longer and thus more specific mask compared to the original legitimate prefix. The further attack method is called a one-hop prefix hijack. The attacker modifies the AS path and propagates a fake direct connection to the victim AS [3].

Nevertheless, several techniques have been designed to prevent these types of attacks, i.e., BGP Sec [4] and RPKI [5]. However, these techniques are not widely adopted yet, and they are rising slowly because they can be challenging to deploy. Issues include the requirement for modification of the routing protocols or adding additional network resources [3]. Another way to increase the security may be the usage of blockchain. Being decentralized, trustless, accessible, and transparent, can help to achieve a higher level of protection.

Blockchain technology is similar to a distributed database with some important design differences. It maintains a consistent copy of a particular dataset across several nodes. It is a fully decentralized environment. Every new record has a reference on the previous record added to the blockchain. The record is called a block and has the reference on the previous block. It is calculated as a hash value from the header fields of the previous block. New blocks in the blockchain network are created by miners, who collect all transactions from the network, and after having a certain amount of them, they turn it into the new block. This new block becomes a candidate to be added to the blockchain. The process where nodes must agree with the new block before it is added into the chain is called consensus. If most of the nodes in the consensus agree with the candidate, it will be added to the chain [6]. The consensual algorithms are, i.e., Proof of Work, Proof of Stake, or Practical Byzantine Fault Tolerant. The selected consensual algorithm affects the time of reaching a consensus. Each of these algorithms requires a different amount of agreeing with nodes, and each of the algorithms tolerates different amounts of malicious nodes without any affect on the final agreement. The blockchain can be either private, controlled by a centralized authority or public that allows anyone to participate in the network [7].

We were inspired by the model [8] for managing network devices and model BGPCoin [9] which designs a reliable way for the origin authentication of the resources in the advertisements. It performs and audit BGP resource assignments on the blockchain network, which is resistant against modifications. It ensures security against an attack such as IP prefix hijacking.

We propose a solution to increase the level of BGP routing security with blockchain. In the proposed architecture, we not only verify received BGP advertisements like BGPCoin but we also avoid learning fake prefixes. Our contributions are:Addition of a method for routers comprising an AS to learn all prefixes assigned to the AS automatically. This way we inhibit routers from advertising prefixes with their AS origin if those prefixes are not assigned to the AS.We implement proof-of-concept of our architecture in the public blockchain Ropsten test network. We use measured results to assess the time required to update routing information and cost in Gas based on the number of prefixes per transaction.

In this section, we described attacks on BGP and their possible prevention using blockchain. The rest of this paper has the following structure. In Section 2, we focus on similar solutions. In Section 3, we describe the design of our architecture. Our results are presented in Section 4 and are further discussed in Section 5. Conclusions and future work are explained in Section 6.

## 2. State of the Art

Management of traditional networks is essential for reliable provisioning of services. BGP is not sufficiently secure, and it enables some attacks on the AS path.

In the section, we review notable results that increased the security of communication between ASes. Towards blockchain-based security in BGP Xing et al. [9] proposed BGPcoin, a BGP framework that is created by a set of smart contracts in the network Ethereum. BGPcoin provides a reliable and transparent allocation of sources such as IP prefixes and AS numbers. It performs and checks sources assignment using blockchain resistant to tampering. Sadd et al. [10] also designed RouteCoin model; bi-hierarchical blockchain system. RouteCoin consists of two chains. The global chain is shared among subgroups, and the subgroup chain is shared among ASes. Every AS is placed to subgroup according to its geographical position. They aimed to increase protection for BGP prefix hijacking. They also modified the protocol to achieve faster consensus. Hari et al. [11] introduce the use of a blockchain mechanism to secure BGP and DNS infrastructure. They describe the current problems of RPKI in the decentralized architecture of ASes. They also designed the format of the transaction to save IP prefixes for ASes. However, their solution did not provide an architecture or model that can be implemented among ASes. They also referred to the problem of blockchain scalability. Ma et al. [12] proposed a blockchain-based framework called BGP-LS Chain. It connects multiple ASes and allows us to share bandwidth and delay information of ASes and inter-domain links. They implemented the mechanism on the verification of shared data with smart help contracts. They also introduced an AS credit rating mechanism to evaluate the credibility of every AS. Owners of ASes are forced to advertise the right information. Otherwise, their credit rating is lower, and in the end, less traffic is routed through them. Moreover, they describe research to secure BGP without the use of blockchain.

A few studies utilize an approach based on BGP and Deep learning to detect IP hijack. Shapira et al. in [13] introduce a method using deep learning. They tested their method on past attacks between 2008 and 2018. They were successful in detection of 32 of 48 attacks. In [14], the authors explored identifying conflicts by using their detector, which learned characteristics of good route updates. There are few results in this area that would allow conclusions to be drawn. In future work, it is also suitable to investigate other methods using deep learning such as neural network that are capable of exploiting the memory in the system for enhanced forecasting capability [15].

In [8], the authors introduced distributed architecture for storing configuration files of the network devices. Therefore the device does not need to rely on a central point in the network, but it can obtain the current configuration file from any node in the blockchain network. The advantage of this architecture is that admin can track changes that happened in the past. Every modification of the configuration file of any device is stored in the blockchain. It is simplified to find out when the configuration file was changed and what was modified.

The process of updating a device consists of several steps. First, the network administrator logs in to a blockchain node. The authentication method can be a password or a certificate. Then, he downloads the current configuration file of the device and modifies it. Then the file is checked for syntax errors. If everything is correct, the file is submitted to the network as a transaction. The blockchain node captures this transaction and adds it to the block which is currently being created. If the block passes the consensus process, it will be added to the chain. Now, the file is saved in the blockchain. The network device regularly checks for the new blocks in the chain. If it finds a new block, it checks if the block is addressed to it. Then, it downloads the block and extracts the configuration file. The network device updates its configuration file. Finally, it records acknowledgement about the result of the update into the chain. The authors later extended their work in [16] where their architecture was applied in the IoT environment. The whole architecture is shown in Figure 1. Our objective is to modify this architecture for use with BGP ASes. We implement public blockchain into their architecture and deploy a set of smart contracts for the management of AS border routers. We want to reach the transparent configuration of BGP routers, non-repudiation of these routers configuration history and avoid misconfigurations.

## 3. Design

The whole architecture is shown in Figure 2. Our design connects the two modules. The one describes how to distribute configuration file securely into the router over the blockchain network. The second architecture proposes a reliable way for authentication assigned BGP sources against IP prefix hijacking. Although, if the border router receives BGP update message from a neighbour, it verifies Route Origin Authorisation (ROA) information in blockchain for every IP prefix. This solution protects us against accepting unauthorized IP prefixes, which might be used in IP prefix hijacking. However, it does not any inhibit the spread of these IP prefixes. Therefore, we introduce the architecture focused on adverting only authorized prefixes in BGP update messages by routers. We propose the following solution: the router downloads IP prefixes from the blockchain network which it inserts into an update message. Before this step, the administrator must upload a list of IP prefixes for the router. Nodes in blockchain subsequently verify it. Also, it checks if there is the valid entry stored in blockchain for every prefix of the list. This architecture includes only prefixes that are verified to be assigned to the AS of the router. Thus, routers add to their configuration only valid IP prefixes from the blockchain network and will advertise through BGP only legitimate information. This approach protects against administrator accidental misconfiguration or malicious appropriation of IP prefix of other AS. Nevertheless, it does not affect the dissemination of prefixes that have been learned from neighbouring ASes.

The architecture consists of a set of smart contracts that represent participants and their operations in the RPKI model. Additionally, there is a smart contract for storing ROA entries and list of prefixes (hereinafter referred to as ’subconfig’) for the router. The process flow is outlined as follows (see Figure 3):RIR *R_x_* allocates prefixes.*R_x_* registers to BGPMC, it is a main smart contract where are stored ROA entries and subconfiguration of the router.*L_x_* requests to create LIR contract.*R_x_* allocates prefixes for LIR which LIR will assign ISPes.*R_x_* adds LIR to ACL in BGPMC smart contract.*ISP_x_* requests LIR to assign an IP prefix to it.The prefix is verified to see if ROA record already exists with this IP prefix.Verified IP prefix is recorded in a new block together with mask and max length as ROA entry in the blockchain.*ISP_x_* adds a new subconfiguration for border router.For every prefix of the subconfiguration is checked ROA entry in the blockchain.Verified subconfiguration is uploaded to the blockchain.Affected router downloads the block and loads the new subconfiguration.

### 3.1. Operations

In the section above, we say that architecture consists of a set of smart contracts. It allows entities to manage their prefixes and subconfigurations. Also, it decreases the risk of the spread of prefixes with false origin. It allows the router to advert prefixes with its AS if they are in fact owned by its AS. There are three types of smart contracts:BGP Prefixes Manager Contract (BGPPMC) is the leading smart contract that is common to all participants. It is built by the central authority of IANA. It keeps ROA entries and subconfigurations of routers in the one place.RIR contract is designed for Regional Internet registry. It is responsible for allocation of IP prefixes and create smart contracts to LIR.LIR contract is intended for local/national Internet registry. They lease prefixes with the right to create ROA entries that are bound on their resources.

RIR add/del prefix starts with a smart contract receiving a request to add/del a prefix. It contains fields such as prefix and mask. These operations describe the process of allocation/deallocation prefixes.

RIR createLIRContract request obtains fields such as the public address of LIR wallet, name LIR and array of prefixes. It verifies if there is already created a smart contract for LIR address. If successful, a prefixes verification process is performed. This process searches if some of the prefixes are already assigned to LIR. Finally, it checks if all prefixes are in the pool of allocated prefixes. If this process is successful, it is created a LIR smart contract. The RIR, that initialized create operation is notified of the result and obtains the address of LIR smart contract. The operation is logged to the blockchain.

LIR isPrefixMe operation verifies if the prefix is allocated for LIR smart contract. It ensures that LIR can not create ROA entry with prefix which is not allocated for it.

LIR setROA/removeROA/addISP are operations which call external functions of BGPPMC smart contract.

BGPPMC addROA starts with smart contract receiving a request from LIR smart contract. The request contains arguments such as AS number, prefix, max length of the prefix and validity period. It subsequently validates format of the IP prefix. If successful, it is performed to verification if there is ROA entry for the prefix. Once found of the ROA entry, the process is failed, otherwise network mask and max length of the prefix are compared. If the mask is less than or equal to the max length, it finally verifies if the prefix is owned by LIR. After a successful verification process, the new ROA record is created and the record validity period is set. The action is also logged to the blockchain. BGPPMC delROA is a reversal operation with one change. The argument of the validity period is left out.

BGPPMC getROA request contains arguments such as prefix and AS number. Result of the request obtains ROA entry for the IP prefix.

BGPPMC addISP operation adds the new ISP to ACL. It has arguments such as AS number and public address of the ISP wallet.

BGPPMC addDeviceConfiguration request contains field such as AS number, ID router and an array of prefixes. It adds new subconfiguration for the router to the blockchain if the verification process is successful. At first, it checks if the IP prefix matches with some record of ACL. If successful, the verification of the AS number is performed. It verifies if the source address of request leases the AS number. Finally, it searches a valid ROA entry for the IP prefix. If ROA entry exists, subconfiguration will be added to the blockchain. The affected router is notified of the new subconfiguration and operation is also logged. The data structure of the subconfiguration is shown in Figure 4. BGPPMC delDeviceConfiguration is a reversal operation with one changed. It omits argument with an array of prefixes.

Arguments for BGPPMC getDeviceConfiguration are AS number and ID router. This operation reads current subconfiguration for the ID router, and the result is sent to the requestor.

### 3.2. Known Limitations

Our design has the same security limitations as the original work [16]. Further, the number of prefixes in the block is limited by the max size of the block. Max number of the prefixes is currently 384 per transaction. This restriction can be eliminated by adding off-chain storage such as an external database. Subconfiguration with unlimited number of IP prefixes will be stored to off-chain storage. Hash of the subconfiguration will subsequently be uploaded to the blockchain.

## 4. Results

We developed a proof-of-concept implementation of the designed architecture using the Ropsten Ethereum test network [17]. We have chosen the Ropsten network mainly because BGP information is publicly available. A set of smart contracts outlined in the previous section was implemented in Solidity language. As an access point to the blockchain network, we use an Infura node [18], which provides scalable API access to the Ethereum. Actors interact with smart contracts using wrappers we created with the use of Web3.py library [19].

We first researched the process of allocating and assigning BGP resources with RPKI. On base this process, we found out how to correctly interpret relationships between our smart contracts. Three virtual machines with OS Ubuntu Server 20.04 LTS [20] created our test environment. Each of these virtual machines represented one of the participants, such as RIR, LIR and ISP. These virtual machines were deployed in various geographical locations. We created scripts for each participant. One for the RIRs is used for the assignment of the IP prefixes and for creating the LIR smart contract. LIR script was made for creating and removing ROA entries. Finally, there is an ISP script that manages subconfigurations of the routers.

The first test scenario consisted of 100 measurements of time used to set the new ROA entry in the Ropsten network. These show that the average duration of setting ROA is 35 s. The median of the same operation is 31.11 s. The graph in Figure 5 shows results and the time distribution from these time measurements. The measured values range from 21.6 to 63.2 s.

The second scenario was performed by making 100 measurements of time used to apply a new subconfiguration onto the ISP router. It is the duration of uploading a new subconfiguration to downloading load the new subconfiguration to the router. The duration average of this operation is 37.5 s. The minimal time of setting subconfiguration is 23 s. The maximum observed time is 64 s, and the median value is 34.7 s. Figure 6 depicts the results, with the distribution graph of measured values on the left side. The graph of duration time of downloading and loading the subconfiguration is shown on the right.

The third scenario examines the gas consumption. The gas unit measures the amount of work required to perform a specific operation on the network. The unit of gas is paid in Gwei. The term Gwei refers to a small denomination of Ethereum which can be an analogy with cents in the Euro [21]. We assume that if we increase the number of prefixes in one transaction, the transaction cost will rise. Figure 7 displays the gas consumption related to the number of prefixes grouped into one transaction. The graphs represent the gas unit’s consumption per one transaction and one prefix, respectively.

## 5. Discussion

The results shown in the previous section prove that it is possible to use this architecture to manage AS border routers and increase BGP security. It is possible to control any AS border router connected to our architecture remotely and automatically. In our use case, the main objective is the management of the configuration of prefixes that verifiably belong to the AS. Routers can then advertise these prefixes to peers. If it is necessary to add or delete prefixes, a verified user adjusts the relevant records stored in the blockchain. Local routers then download new prefixes and modify their local BGP configuration to reflect the adjustments. This means that part of the configuration of an AS border router can be decentralized, distributed, immutable, and provenance of all prefixes configurated and announced are known. These properties can help defend against BGP hijacking, and reduce the number of fraudulent BGP announcements and local misconfigurations.

The first and second scenarios endeavour to evaluate the performance of the proposed architecture using time for modifications to distribute. Looking at the results, we find out that the current block time [22] influences the operation’s duration. Furthermore, our transaction may not be mined in the first block after uploading it to the pool of pending transactions. It may be mined in one of the subsequent blocks. We can influence this allocation into blocks and motivate miners by fine-tuning the gas price of our transaction. Therefore, we use the setting “fast_gas_price_strategy” from library Web3.py, which ensures the mining of the transaction within 60 s. Note that downloading the subconfiguration to routers from blockchain has little influence on the duration time of the whole operation.

The final scenario focuses on the gas consumption of the operation uploading of the subconfiguration. Gas consumption per transaction slowly increases in an approximately linear fashion with raising the number of prefixes in the subconfig. On the other hand, the most expensive operation is with only one prefix. At first, the cost sharply falls per prefix, and then it is approximately constant. Therefore, it is cheaper to aggregate multiple prefixes into one transaction. The average gas consumption is 3,710,704 gas units. The current price of 1 gas unit is 26 gwei [23]. Therefore, the cost of the operation is equal to 22.38 dollars.

The use of the blockchain in any architectures has some disadvantages. The verification of a transaction in a network requires to agree with a majority of participants. Most of the public blockchain networks use consensus algorithms that consume the amount of computational power and energy. It has negatively affected the environment and sustainability values [24]. We believe that a fundamental improvement will come with the release of Ethereum 2.0 [25]. It replaces consensual algorithm proof of work by proof of stake that reduces consumption energy [26]. The other way to support sustainable energy management is by increasingly using alternative energy resources by miners such as solar energy [24].

The blockchain size can grow huge over time. Currently, it requires over 290 GB of storage [27]. The block size is also limited; therefore, it can restrictive to store original data into the blockchain. It is practical to use decentralized storage systems such as the InterPlanetary File System with blockchain. It stores the hash of the original data in the chain, instead of the data itself [28,29].

Despite these limitations, it is beneficial to use blockchain in our architecture and others. It provides high security for the architecture, and there is already an infrastructure for deploying our application.

## 6. Materials and Methods

Our source code is available at GitLab [30]. The repository includes smart contracts such as RIRManagerContract, LIRContract and BGPPMC. There are also wrapper scripts for participants of the system. The contracts addressed in the Ropsten network are as follows:0xd57a1aAD0fD9f1E27Fadb659136A868D360b42510x5bF128C371BCD858e398F1432701c38938bCe0660xf8F78b24819b0219aF51905DE88Adb39a560a296

## 7. Conclusions and Future work

As discussed in the introduction, BGP has some security deficiencies. It is possible to hijack prefixes to influence the routing of traffic. To increase resistance against this attack, we modified the original architecture for use with edge routers of ASes. Our modified architecture provides decentralized and immutable storage for prefixes allocated to an AS. Finally, we have implemented a set of smart contracts to manage BGP configurations for edge routers of the AS.

Our solution enables verification of the prefixes before deployment to routers and limits the number of fraudulent BGP origin announcements. Moreover, it keeps the irrefutable history of all changes through blockchain. Related works employing blockchain concentrate on reactive verification of prefix information in BGP update packets when they arrive. Our solution’s key novelty is that we focus on the proactive prevention of fraudulent BGP origin announcements in addition to reactive BGP origin AS validation. We achieve this by deploying verifiably correct configurations to routers in origin AS.

As future work, we want to focus our research on a different environment for this architecture. We want to test this architecture in more extensive topologies to obtain results more relevant to a large scale of the Internet. Further advances should target reducing attacks on BGP by following the AS paths and take into account an extended list of BGP attributes. A possible avenue of improvement is to add information about links to other peers to the transaction. We also aim to enhance the periodic checking of AS updates in the current implementation. This process could be replaced by sending notification messages when relevant changes have been propagated through the blockchain. The last improvement to the architecture we are aware of is a more efficient consensus algorithm [31]. Current implementation uses Proof of Work. If we replaced it, we could decrease the duration time of the uploading subconfiguration.

## Figures and Tables

**Figure 1 sensors-20-04482-f001:**
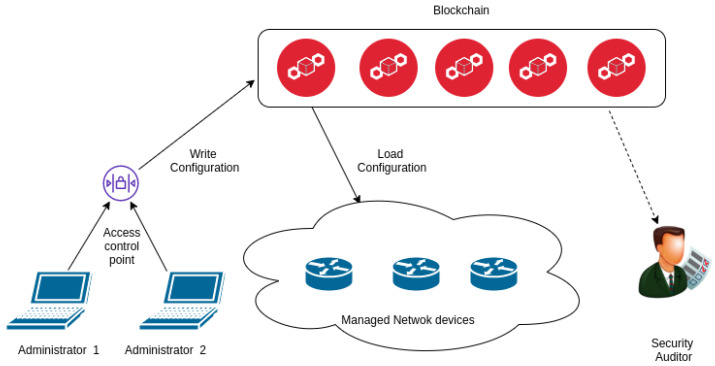
Blockchain architecture for management network devices. Reproduced from [8] and permission obtained from IEEEE.

**Figure 2 sensors-20-04482-f002:**
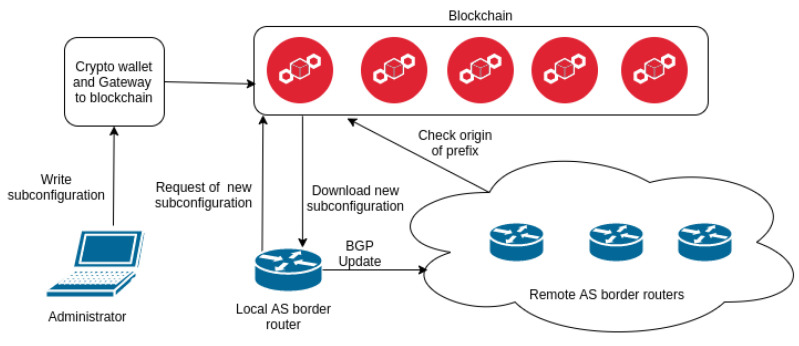
Design architecture.

**Figure 3 sensors-20-04482-f003:**
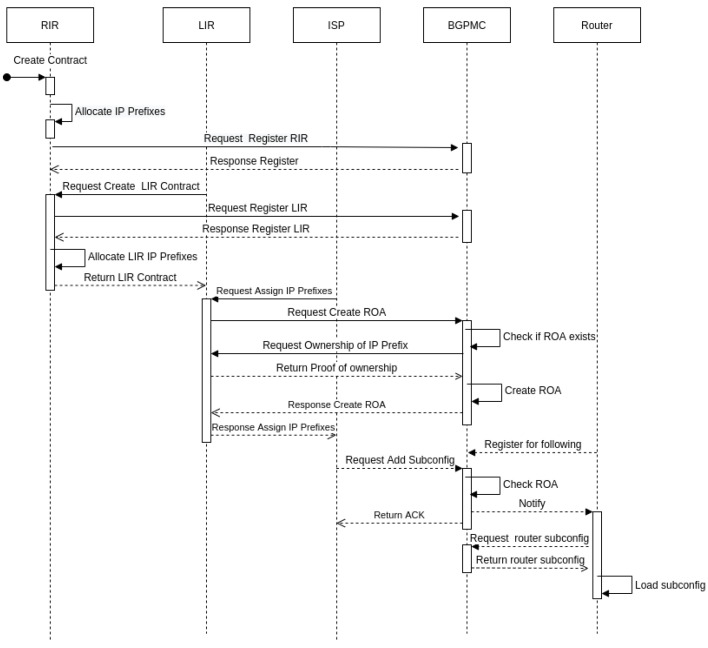
Management sequence diagram.

**Figure 4 sensors-20-04482-f004:**
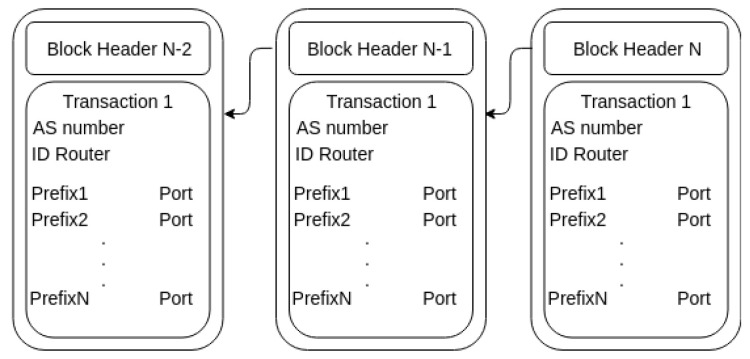
Data structure for block.

**Figure 5 sensors-20-04482-f005:**
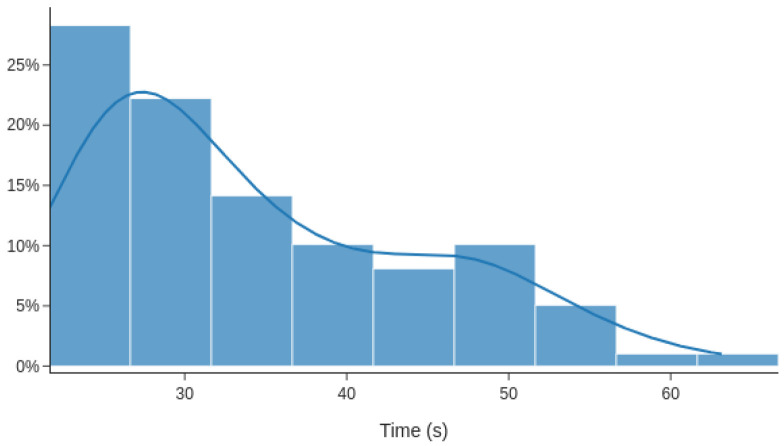
Time duration distribution of setting Route Origin Authorisation (ROA).

**Figure 6 sensors-20-04482-f006:**
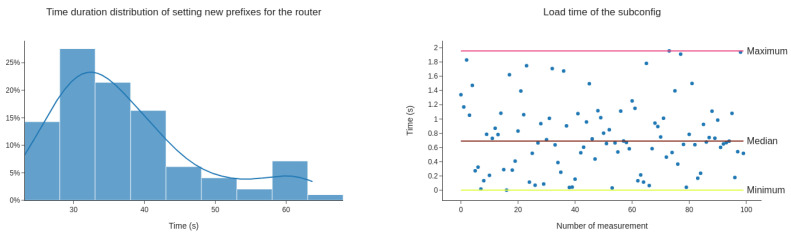
Time duration distribution of setting new prefixes for the router

**Figure 7 sensors-20-04482-f007:**
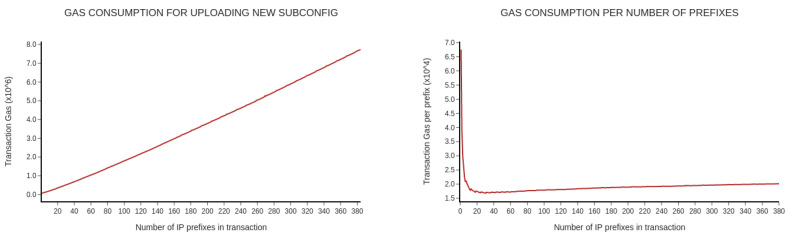
Gas consumption.

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
