# Peer review of "Enhancing Border Gateway Protocol Security Using Public Blockchain"

_sensors, 2020, doi:10.3390/s20164482_

Round 1

Reviewer 1 Report

I like the fact that the paper is based in a reproducible open-source approach, and that the code is available to everyone. However, please address the following:

  • In title, better use Border Gateway Protocol than the acronym.
  • Relate the topic to novel methods with artificial intelligence. AI can be used to monitor anomalous behaviour such as attacks. For example, reference AI for detecting anomalies with RNNs: “Deep Recurrent Entropy Adaptive Model for System Reliability Monitoring” 1109/TII.2020.3007152, and with autoencoders.
  • Add Ubuntu to references
  • What are the disadvantages in terms of computational cost, storage capacity and energy consumption the use of a blockchain ledger? That should be more carefully explained in the paper.
  • Figure 6b would be more interesting with more cases and error bars.

Improve English grammar and paper style. For example:

  • ‘use a public blockchain Ethereum…’ -> ‘use the public Blockchain Ethereum…’
  • ‘In this section, we describe attacks on BGP and their possible prevention using blockchain.’ Why the contents of the section are described at the end?
  • The authors of [9] -> In [9], the authors…

Author Response

Point 1: Relate the topic to novel methods with artificial intelligence. AI can be used to monitor anomalous behaviour such as attacks. For example, reference AI for detecting anomalies with RNNs: “Deep Recurrent Entropy Adaptive Model for System Reliability Monitoring” 1109/TII.2020.3007152, and with autoencoders.

Response 1: We have added information about BGP and Deep learning in the third paragraph in the "State of the art": "A few studies utilize an approach based on BGP and Deep learning to detect ..."

Point 2: Add Ubuntu to references

Response 2: We have added the reference to OS Ubuntu in the revised manuscript.

Point 3:What are the disadvantages in terms of computational cost, storage capacity and energy consumption the use of a blockchain ledger? That should be more carefully explained in the paper.

Response 3: In the section "Discussion", we have added two paragraphs that explains the disadvantages of using blockchain in architecture; it starts from: "The use of the blockchain in any architectures have some ...". These disadvantages are the same for any application that is base on the blockchain.

Point 4: Figure 6b would be more interesting with more cases and error bars.

Response 4: We have changed the chart's mode from "lines" to "markers" because these measurements are independent of each other. We have also added the median, maximum, and minimum line.

Point 5: Improve English grammar and paper style.

Response 5: Our English editor has edited the paper.

Point 6: ‘In this section, we describe attacks on BGP and their possible prevention using blockchain.’ Why the contents of the section are described at the end?

Response 6: There was a mistake. We should have used the past simple. We have fixed it.

Reviewer 2 Report

In the present paper the authors propose a solution to increase the level of BGP routing security with the use of blockchain, computing time and  costs.

The paper has original steps that are inventive toward the solution of BGP hijacking.

I would suggest to the authors to add some hints, references related to the scopes of the journal to foster the applicability of their proposal to the sensors aims and scopes

For the sake of clarity the authors, before speaking of gas token, should define what it is and related gwei expecially for the comprehension of the conclusions section. The paper is focused on costs of transactions but no mention htere were about energy costs and sustainability of the proposed model. Please check in the mdpi journals those dealing with sustainability of blockchain technology and add references to this important issue.

Author Response

Point 1: I would suggest to the authors to add some hints, references related to the scopes of the journal to foster the applicability of their proposal to the sensors aims and scopes.

Response 1: We have added information about BGP and Deep learning in the third paragraph in the "State of the art": "A few studies utilize an approach based on BGP and Deep learning to detect ...". In the section "Discussion," we have also added two paragraphs that explains the disadvantages of using blockchain in architecture with references related to the MDPI journals.

Point 2: For the sake of clarity the authors, before speaking of gas token, should define what it is and related gwei expecially for the comprehension of the conclusions section.

Response 2: We have explained what is gas unit and gwei in the final paragraph in the "Results".

Point 3: The paper is focused on costs of transactions but no mention htere were about energy costs and sustainability of the proposed model. Please check in the mdpi journals those dealing with sustainability of blockchain technology and add references to this important issue.

Response 3: We have described sustainability in the last three paragraphs in the "Discussion", but this paper's main aim is not about the sustainability of the design. Therefore, we do not very devote to effective energy consumption or the minimum size of storage in this paper, but it is a good idea for future work. Our main aim is, decrease the spread of fake IP prefixes. We have added references related to sustainability - open access journal.

Round 2

Reviewer 1 Report

The Authors have addressed all the reviewer's comments, thus I recommend the paper for publication.

Reviewer 2 Report

The paper in its present form is worth of publication

This manuscript is a resubmission of an earlier submission. The following is a list of the peer review reports and author responses from that submission.